OBC-YOLOv8: an improved road damage detection model based on YOLOv8

Zhang Shizheng
Liu Zhihao 1062698149@qq.com
Wang Kunpeng
Huang Wanwei
Li Pu
Software Engineering College, Zhengzhou University of Light Industry , Zhengzhou, Henan , China
Yang Jiachen
Electronic publication date: 2025 Jan 7
Publication date: 2025
Volume: 11
Electronic Location ID: e2593
Received 2024 May 21; Accepted 2024 Nov 18
Copyright: © 2025 Zhang et al.
Copyright year: 2025
Copyright holder: Zhang et al.
License: This is an open access article distributed under the terms of the Creative Commons Attribution License, which permits unrestricted use, distribution, reproduction and adaptation in any medium and for any purpose provided that it is properly attributed. For attribution, the original author(s), title, publication source (PeerJ Computer Science) and either DOI or URL of the article must be cited.
License URL: https://creativecommons.org/licenses/by/4.0/

Keywords: Road damage detection, Deep-learning, YOLOv8, Attention mechanism, Dynamic convolution

Funding: Henan Province 24B520038 The work is supported by the key scientific research projects in higher education institutions in Henan Province (24B520038). The funders had no role in study design, data collection and analysis, decision to publish, or preparation of the manuscript.

==============================
Effective and efficient detection of pavement distress is very important for the normal use and maintenance of roads. To achieve this goal, a new road damage detection method based on YOLOv8 is proposed in this article. Firstly, omni-dimensional dynamic convolution (ODConv) block is employed to better grasp the complex and diverse features of damage objects by making dynamic adjustment according to the features of input images. Secondly, to extract the global and local feature information simultaneously to better improve the feature extraction ability of the model, BoTNet is added to the end of the backbone, which can combine the advantages of convolutional neural network (CNN) and Transformer. Finally, the coordinate attention mechanism (CA) is incorporated into the Neck section to make more accurate speculations and enhance detection accuracy further which can effectively mitigate irrelevant feature interference. The new proposed model is named OBC-YOLOv8 and the experimental results on the RDD2022-China dataset demonstrate its superiority compared with baselines, with 1.8% and 1.6% increases in mean average precision 50 (mAP@0.5) and F1-score, respectively.

Introduction

With the rapid development of social economy, the demand for logistics is increasing and the role of traffic road infrastructure is becoming more and more important. However, the inclement weather, natural disasters, traffic accidents and other factors cause great damage to the roads and threaten the safety of people’s lives and property. So, the road surface must be maintained in a reliable and cost-effective manner. Timely road condition monitoring is also required (Koch et al., 2015). Effective and efficient road damage detection can achieve the goal of finding the road damages early and perform road maintenance in time, ensuring the safety and reliability of traffic roads.

The existing road damage detection technologies can be generally divided into two types: one is the traditional detection technology and the other is the deep learning-based detection technology (Luo et al., 2023). Traditional road damage detection methods are mostly based on image processing and image segmentation techniques using manual or traditional classifiers. For example, Wang, Sha & Sun (2010) proposed a new road crack classification method based on chain code and obtained the crack skeleton in the road image by refining filtering, using crack chain code tracking and crack classification. Zhou, Huang & Chiang (2005) proposed a classification method of pavement distress based on wavelet transform and Radon transform. These traditional methods frequently necessitate intricate image preprocessing procedures and heavily rely on manual feature extraction, ultimately leading to suboptimal performance outcomes.

In recent years, with the development of deep learning, more and more researchers leverage different deep-learning based models for road damage detection. These methods are mainly categorized into two types: one is a two-stage model and the other is a one-stage model (Huang et al., 2023). The two-stage model first generates a series of candidate boxes as samples and then classifies samples through convolutional neural network. The commonly used two-stage models include Fast region-based convolutional neural network (R-CNN) (Girshick, 2015), region-based fully convolutional networks (R-FCN) (Dai et al., 2016), Mask R-CNN (He et al., 2017), and CoupleNet (Tay, Tuan & Hui, 2018). For example, Fan et al. (2018) proposed a supervision method by extracting small blocks from crack images as input and generating a large training database to train convolutional neural networks (CNN) which achieves the goal of automatic pavement disease detection. But due to the limitations of the labels judged by human experts, thinner crack labels lead to thinner crack outputs. Li et al. (2020) proposed a novel method using deep CNN to automatically classify image patches cropped from 3D pavement images. However, the classification accuracy is affected by the size of receptive field, and the smaller the receptive field, the longer the training time. Tang et al. (2021) divided the pavement highlights into patches and then used the expectation maximization inspired patch label distillation (EMIPLD) strategy to infer the patch labels from the pavement images for pavement disease classification. Pham, Pham & Dang (2020) used Detectron2 and Faster R-CNN to find a general model suitable for different areas, achieving automatic detection and classification of road damage. Fan & Liu (2019) proposed a novel road damage detection algorithm based on unsupervised disparity map segmentation. Hacıefendioğlu & Başağa (2022) focused on detecting cracks in concrete roads for various shooting, weather conditions and illumination levels using a pre-trained Faster R-CNN. They found that under bad weather or insufficient light conditions, the model’s ability to detect cracks dropped significantly. Shim et al. (2022) developed a new sensor technology that can detect road damage. The proposed technology includes a super-resolution and semi-supervised learning method based on a generative adversarial network. Huang et al. (2023) proposed a series of simple yet effective end-to-end deep learning approaches named Weakly Supervised Patch Label Inference Networks (WSPLIN) for efficiently addressing these tasks under various application settings. Its versatility stems from three distinct image sampling methods, enabling it to adapt seamlessly to a wide range of detection tasks. Jin et al. (2022) proposed a deep learning model based on feature points from a local minimum of grayscale. The trained CNN model was used to output the feature vector, which was then input into a support vector machine (SVM) for classification. The smaller size allows it to achieve high precision with little machine time, even at mobile terminals.

The one-stage model is a kind of road damage detection method based on a single stage. In general, the one-stage model regards the detection problem as a single regression problem and directly outputs the category and location information of the damage, usually using some anchored detection methods. The well-known YOLO (You Only Look Once) (Redmon et al., 2016) series, SSD (Single Shot MultiBox Detector) (Liu et al., 2016), and RetinaNet (Lin et al., 2017) fall into this group. In addition, many relevant studies have been conducted. For example, Roy & Bhaduri (2023) proposed a real-time deep learning (DL)-based high-performance damage detection model where DenseNet blocks were integrated with the backbone to preserve and reuse the critical feature information more effectively. Their work constitutes a step towards an accurate and robust automated damage detection system in real-time in-field applications. Arya et al. (2020) mentioned that IMSC applied the test time augmentation (TTA) procedure on test data to improve their model’s robustness. TTA augments the data by using several transformations and obtain good performance in the detection of road damage images in different countries. Ding et al. (2022) achieved the best results in each country’s ranking using the YOLO series model and Faster R-CNN as baselines on the CRDDC dataset, respectively. Rout et al. (2023) proposed a novel approach for improving the performance of pothole detection using low-resolution cameras or low-quality images and video feed. They employed enhanced super resolution generative adversarial networks (ESRGAN) to enhance the images, and integrated this with YOLOv7 for detection, resulting in a notable improvement in the model’s performance for pothole detection. Xiang et al. (2023) proposed an improved road damage detection algorithm, YOLOv5s-DSG. It uses the Ghost module to replace the traditional convolution to reduce the number of model parameters, and introduces the Space-to-depth Conv module to adapt to low-resolution and small object detection tasks. These make the model more lightweight and improve the detection efficiency and accuracy. Xu et al. (2023) proposed a dense multi-scale feature learning Transformer. The cross-shaped attention mechanism is used to extend the perceptual domain of feature extraction and the intensive multi-scale feature learning module is used to integrate local information at different scales. But their study focused solely on the detection of potholes and did not encompass the category of cracks. Compared to traditional methods, deep learning-based methods can often achieve better performances. However, most of these methods pay little attention to the specific features of damage images. With respect to road damages, the shape characteristics of cracks, potholes and other damages are complex. What’s more, the backgrounds of roads are also varied. Adverse weather conditions and poor lighting can significantly impact the accuracy of detection, leading to potential inaccuracies and reduced performance. All these facts indicate that accurate road damage detection is not an easy task. So, it needs to handle the specific features of road damage well.

In this article a novel road damage detection model named OBC-YOLOv8 is presented. The main contributions of this article are listed as follows:

1. Omni-dimensional dynamic convolution (ODConv) is applied to actively adapt to the feature shape in the image and strengthen the feature extraction ability of the model.

2. BoTNet module is added to strengthen the global and local feature information of the feature map.

3. Coordinate attention mechanism (CA) is introduced to suppress invalid features and enrich the effective information in the feature network.

4. Extensive experiments have been conducted to demonstrate the effectiveness and efficiency of OBC-YOLOv8 in road damage detection tasks.

Related work

The two-stage model generally has high detection accuracy, but the speed is usually slow. So, for many real-time detection tasks, the single-stage detection model is chosen alternatively. YOLO series are the well-known one-stage detection models in recent years. YOLOv8 is famous for its excellent object detection performance. For example, Wu & Dong (2023) employed a modified YOLOv8 algorithm for the analysis of optical remote sensing images. To enhance the network’s feature extraction capabilities, an enhanced slight editor fine (SEF) module based on Slight Edition Convolution (SEConv) was proposed to reduce the number of parameters. An efficient multi-scale attention (EMA) mechanism was integrated into the network, forming the spatial pyramid pooling with less FLOPs edition (SPPFE) module. This enables the model to excel in addressing the challenges of multi-scale object detection. Lou et al. (2023) introduced an innovative small-object detection algorithm tailored for unique scenarios, leveraging the foundations of YOLOv8. This approach incorporates a novel downsampling technique that adeptly retains contextual feature information, coupled with a newly devised network architecture aimed at enhancing detection accuracy. Wang et al. (2023) proposed an improved YOLOv8 algorithm for road hazard detection. By integrating the bi-directional feature pyramid network (BiFPN) concept, the model’s parameters, computational load, and overall size were reduced. The introduction of the simplified SPPF (SimSPPF) module optimized the feature pyramid layers, enhancing their speed. Furthermore, the inclusion of large separable kernel attention (LSK-attention) expanded the model’s receptive field, improving the accuracy of road damage detection.

To ensure real-time response for road damage detection, YOLOv8n combines the strengths of prior YOLO models. By enhancing network structure, training strategies, and inference speed, it achieves high detection accuracy and speed. This allows quick identification and localization of road damages in real-world scenarios. For dealing with the diversity of road damage sizes, YOLOv8n utilizes improved multi-scale prediction. This technology predicts on multiple scales, effectively detecting damages of different sizes. In summary, due to its performance, real-time capability, and multi-scale detection, we choose YOLOv8n as our baseline.

YOLOv8n network

The main idea of YOLO series is as follow: at first dividing the image into grids; then employing bounding boxes to predict the class of the object contained in each grid; finally using non-maximum suppression (NMS) algorithm to eliminate overlapping bounding boxes. Similar with Darknet-53, YOLOv8n uses the structure of convolutional modules and residual blocks for feature extraction on the backbone network. The main difference is that YOLOv8n uses the Conv+batch normalization+SiLU (CBS) block and the CSP bottleneck with 2 convolutions structure as the convolution block and the residual block respectively. This enhances the feature extraction capability while the model is lightweight. Then a path aggregation network-feature pyramid network is constructed in the Neck using the cross stage partial networks idea, and the feature maps obtained from multiple up-sampling and down-sampling operations were used for cross-level feature fusion. In the head, the current advanced Decoupled Head structure is used to separate the classification head and detection head. Specifically, the performance of the classification head is evaluated using the binary cross entropy (BCE) loss function, whereas the detection head’s performance is gauged by the bounding box loss (Bbox loss), which incorporates complete intersection of union (CIoU) and distribution focal loss (DFL). The final loss function is composed of the weighted average of these three losses. In terms of the sample matching strategy, YOLOv8n opts for the anchor-free approach, discarding the anchor-based method. The anchor-free model determines positive and negative samples in a more efficient manner, achieving comparable or even superior accuracy to the two-stage anchor-based model, while offering faster processing speeds. The overall structure of the YOLOv8n network is shown in Fig. 1.

Figure 1 Overall network structure of YOLOv8n.

Despite YOLOv8n’s excellent performance in the field of object detection, it still has some shortcomings in road damage detection. When confronted with various types and sizes of road damage targets, the traditional convolutional approach makes the model’s reasoning ability somewhat inadequate. Furthermore, the lack of increased attention to the damage targets leads to weaker feature extraction capabilities of the model. Indeed, addressing these limitations and making improvements to YOLOv8n in the context of road damage detection is highly necessary.

Methods

Overview

YOLOv8n model has certain advantages in the field of target detection. However, road damage detection is a task requiring high detection accuracy and real-time performance, and the original model needs to be further optimized to meet the requirements (Zhao et al., 2024). In this section, we introduce the proposed OBC-YOLOv8 model in detail. The main work consists of three aspects and the basic network structure of OBC-YOLOv8 can be illustrated with Fig. 2. Firstly, to improve the feature extraction capability, omni-dimensional dynamic convolution blocks (ODConv) are introduced by replacing some traditional convolution blocks in backbone of the model. Secondly, to better extract global feature information and local feature information of road damage images, a BoTNet module is added to the end of the backbone module. Finally, to further reduce the interference of irrelevant features and improve the accuracy of model detection, we attach a coordinate attention mechanism (CA) to the last two C2f modules in the Neck part, which can fully use the position information.

Figure 2 The network structure of OBC-YOLOv8.

Omni-dimensional dynamic convolution

Convolution has been widely applied in many deep learning approaches since it can effectively improve their performance. However, traditional convolution operation is generally time-consuming, which only focuses on the convolution kernel dimension. Therefore, YOLOv8n still has insufficient reasoning ability when faced with multiple types and sizes of road damage targets. To tackle the above problem, Li, Zhou & Yao (2022) proposed the omni-dimensional dynamic convolution (ODConv) which adds complementary attention of multiple dimensions to convolution in a parallel way. ODConv can dynamically adjust the shape and size of the convolution kernel based on the characteristics of the input data. This enables it to adaptively adjust its detection strategy based on the characteristics of different road damages, effectively detecting different sizes and types of road damages. In this article we leverage ODConv in our model to provide a tradeoff between performance and efficiency. ODConv can be represented by Eq. (1):

(1) YO=(αW1⊙αf1⊙αc1⊙αs1⊙W1+…+αWn⊙αfn⊙αcn⊙αsn⊙Wn)∗xi

where, αwi is the attention to the convolution kernel Wi, αsi is the attention along the convolution kernel space dimension, αci is the attention along the input channel dimension and αfi is the attention along the output channel dimension. ⊙ represents the multiplication operation along different dimensions of the kernel space. The four kinds of attention are calculated with a multi-head attention module πi(x ). The structure is shown in Fig. 3.

Figure 3 The structure of multi-head attention module πi(x ).

The input x is compressed to a feature vector of length Cin by a channel-level global averaging pooling (GAP) operation, which is then further processed by the fully connected layer (FC) and rectified linear unit (ReLU). The fully connected layer maps the compressed eigenvector to the lower dimensional space with a certain reduction rate. For four head branches, each has an FC layer with the output size of k×k, Cin×1, Cout×1, n×1. Then a SoftMax or Sigmoid function is used to generate normalized attention αsi, αci, αfi, αwi,respectively. These four different types of attention complement each other. They are multiplied by the convolution kernel Wi in order of position, channel, filter, and kernel dimensions, and then accumulated to ensure that the convolution operation differs from all spatial positions, input channels, filters, and kernels of input x, ensuring the capture of rich contextual clues. Benefiting from the enhanced feature extraction ability, ODConv exhibits promising performances compared with dynamic convolution with multiple kernels. What’s more, ODConv has less additional parameters, and is more efficient than dynamic convolution (DyConv) (Chen et al., 2020).

BoTNet model

BoTNet (Srinivas et al., 2021) is an exploration by researchers at Berkeley and Google of the combination of convolutional networks CNN and Transformer which can be illustrated in Fig. 4. It uses a hybrid approach to replace ResNet bottlenecks in visual tasks with a Bottleneck Transformer. In detail, BoTNet uses multi-head self-attention (MHSA) layers to replace 3×3 spatial convolution in the last three Bottleneck blocks in the ResNet framework. This enables the model to better extract more rich and detailed features in the image, and capture and identify various damages on the road. In contrast to YOLOv8n, BoTNet gives the model a global view, focusing on every pixel in the image and considering the dependencies between them. It helps the model to take into account the environment and context information of the whole road while detecting local diseases, so as to improve the accuracy of detection. In general, BoTNet offers superior performance compared to CNN or Transformer alone, demonstrating its effectiveness in road damage detection task.

Figure 4 Comparison between ResNet Bottleneck and BoTNet.

Due to the limited feature subspace, the modeling ability of single-headed attention blocks is very rough, while MHSA can solve the problem well. It projects inputs linearly into multiple feature subspaces and is processed in parallel by multiple independent attention heads. The resulting vector is concatenated and mapped to the final output. The diversity of feature subspace is enriched without extra computational cost. The structure of MHSA is shown in Fig. 5. MHSA mechanism can be presented as follows:

(2) Qi=XWQi,Ki=XWKi,Vi=XWVi,i=1…h

(3) Zi=Attention(Qi,Ki,Vi)=SoftMax(QiKiTdk)Vi,i=1…h

(4) YB=MultiHead(Q,K,V)=C(Z1,Z2,…,Zh)Wo

where X is the input, WQi, WKi and WVi are three different groups of linear metrics, Qi, Ki and Vi are the matrix obtained by linear transformation from the input matrix, and the dimension is the same as the input matrix. QiKiT represents the similarity of Qi and Ki, dk is the scaling factor to prevent gradient vanishing during the backpropagation of the SoftMax function due to the large value of QiKiT when the dimension is too high, SoftMax is a normalized function that maps to V to obtain the correlation of channels. Wo donates the output projected matrix, and C stands for concatenation operation.

Figure 5 The structure of MHSA.

In the OBC-YOLOv8, we introduce the BoTNet, which combines convolution and self-attention mechanism to make up for the effect of convolutional information loss. It can use convolution to effectively learn abstract low-resolution feature maps from images, while using multi-head self-attention mechanism to process and aggregate feature information captured by convolution. It also expands the learning ability of the model for different positions and improves the detection accuracy with a weak amount of computation.

Coordinate attention

There are many kinds of road damage and the backgrounds of roads are also complex, so accurately detecting road damage is a difficult task. YOLOv8n only uses convolution operations for feature extraction which lead to a leakage attention to the damage features. The studies on lightweight networks show that the attention mechanism can bring significant performance improvement to the model, and the wildly used attention mechanisms are SENet (Hu, Shen & Sun, 2018) and CBAM (Woo et al., 2018). The squeeze-and-excitation (SE) attention can get the importance of each channel of the feature map while the convolutional block attention module (CBAM) attention mechanism can combine channel attention with spatial attention to extract position information. However, they all have some disadvantages. The SE attention only considers the encoding of information between channels and neglects the importance of position information, which is actually crucial for many visual tasks that need to capture the object structure. CBAM makes use of position information by reducing the channel dimension of the input tensor and then computing spatial attention using convolutions. However, convolutions can only capture local relations but fail in modeling long-range dependencies that are essential for vision tasks (Hou, Zhou & Feng, 2021).

CA (Hou, Zhou & Feng, 2021) is a coordinate attention mechanism, which is used to encode precise location information into neural networks in order to model channel relationships and long-term dependencies. It consists of two steps: coordinate information embedding and coordinate attention generation. The structure of CA is shown in Fig. 6. In the coordinate information embedding step, position information is embedded in the input feature map. In the coordinate attention generation step, the position information is used to generate the attention map and applied to the input feature map to emphasize the representation of interest. The advantage of CA is the ability to participate in modeling large areas in mobile networks and to avoid significant computational overhead.

Figure 6 The structure of CA.

C is the number of channels; H is the height; W is the width; r is a hyperparameter used to adjust the model’s attention to information at different locations.

CA performs feature aggregation on input along the two spatial directions of width and height firstly. Through the global average pooling, we can get feature maps in horizontal and vertical directions with the size of C×H×W and C×1×W, respectively. Then the two feature maps are concatenated and sent to a 1×1 convolution conversion function to obtain the intermediate feature graph. In this way, an intermediate feature containing horizontal and vertical spatial information can be obtained. Then the feature is divided into two independent features and transformed using two more convolution and sigmoid functions. Finally, the two outputs are combined into a weight matrix, which is used to calculate the output of the module. The final output of CA is represented by:

(5) YC=x∗σ(Fh(δ(F1[αh,αw])))∗σ(Fw(δ(F1[αh,αw])))

where αh and αw represent the output of the average pooling in the horizontal and vertical directions, [.,.] is a concatenation operation, F1 means the 1×1 convolution for dimensionality reduction transformation and is a nonlinear activation function, Fh and Fw represent the convolution in height and width directions, σ represents the H-Sigmoid activation function.

CA attention mechanism is introduced into OBC-YOLOv8 to guide the network to pay more attention to the area of damage image. In this way the influence of irrelevant background information can be reduced and the detection accuracy can be improved meanwhile. Furthermore, the CA attention module boasts a small parameter count and doesn’t introduce significant computational overhead.

Loss function

The loss function measures the difference between the predicted value and the real value of the model and is crucial to the training and detection performance of the model. In this article, the loss function of OBC-YOLOv8 consists of several parts, including classification loss and bounding box regression loss.

The classification loss function of OBC-YOLOv8 is BCE Loss, which measures the difference between the probability distribution predicted by the model and the actual label. BCE loss makes a “Yes or No” judgment for each class and outputs the confidence level. The formula of BCE can be described as follows:

(6) LBCE=−1N∑i=1N[yi⋅log(pi)+(1−yi)⋅log(1−pi)]

where yi represents the class and pi represents the confidence level of the class.

OBC-YOLOv8 adopts DFL and CIoU to represent the bounding box regression loss, which aims to measure the overlap of detection boxes more accurately, so as to improve the ability of the model to locate the target object. In the task of road damage detection, the frequency of different types of damage in the image is often different, which leads to the problem of unbalanced category. DFL is designed to solve this problem. DFL pays more attention to those samples that are difficult to classify and rare classes by adjusting the loss weights, which can be described as follows:

(7) LDFL=−((yi+1−y)log(Si)+(y−yi)log(Si+1))

(8) Si=yi+1−yyi+1−yi,Si+1=y−yiyi+1−yi

where y represents the mapping value from the center to the boundary, yi and yi+1 represent the integrated and near predicted values of y, respectively. Specifically, DFL will dynamically adjust the loss weights of different classes according to the distribution of each class, so that the model can handle samples of different classes more balanced in the training process. In bounding box regression loss, the overlap between the predicted and true boxes is an important evaluation metric. CIoU loss function is an efficient and accurate optimization method for target detection and can improve the accuracy and performance of target detection by taking multiple factors into account to evaluate the difference between the predicted and real boxes. CIoU can be presented by:

(9) LC=1−IoU+d2c2+αv

(10) v=4π2(arctanwgthgt−arctanwh)2,α=v(1−IoU)+v

where d is the Euclidean distance between the predicted box and the real box, and c is the diagonal length of the smallest box surrounding the two boxes. The variables w, h and wgt, hgt represent the height and width of the predicted box and the real box, respectively. The variables v is used to measure the consistency of the relative proportions of the two boxes, and α is the weight coefficient. The complete intersection of union (IoU) is the intersection of the predicted box and the real box.

The final loss function is composed of the weighted average of these three losses:

(11) L=wbLBCE+wdLDFL+wcLC

where wb, wd and wc are the weight coefficients of the three loss functions, respectively. During the training process, the model will continuously improve the accuracy of the model prediction by minimizing the value of the loss function. It ensures that the model can have a clear optimization direction in the training process, so as to achieve better detection results in practical applications.

Experiments

In this section, we introduce the experimental configuration, experimental parameters and experimental results of this study in detail.

Data and setup

Experimental environment

All experiments are conducted on an Ubuntu 20.04.5 LTS (64-bit) system with an Intel (R) Core (TM) i5-13600KF CPU and an NVIDIA RTX GPU 4080 (16 GB). The development language of this model is mainly Python, using the open-source deep learning framework PyTorch as the network framework. CUDA12.2 is used to accelerate the training.

Throughout the training phase of all experiments, a consistent set of hyperparameters was rigorously applied to ensure uniformity and comparability. Table 1 provides a comprehensive overview of the exact hyperparameters employed during the training process.

Table 1 Hyperparameters configuration.

Hyperparameters	Value	
Learning rate	0.01	
Image size	640 * 640	
Weight decay	0.0005	
Batch size	16	
Momentum	0.937	
Epochs	300	

Data enhancement

In the field of deep learning, especially in object detection tasks, the importance of data enhancement in model training cannot be ignored. As an effective technical means, it significantly improves the generalization ability and robustness of models by increasing the diversity and complexity of training data sets.

In our experiment, we also used some data enhancement methods of YOLOv8n to help train the model. Table 2 shows the parameter configurations of these methods.

Table 2 Data enhancement configuration.

Methods	Value	
HSV_h	0.015	
HSV_s	0.7	
HSV_v	0.4	
Translate	0.1	
Scale	0.5	
Fliplr	0.5	
Mosaic	1.0	

The HSV approach introduces color variability by adjusting the hue, saturation, and value of images, simulating diverse environmental conditions and helping the model perform well under various lighting conditions. The translate method shifts the image slightly in both horizontal and vertical directions, which is beneficial for learning to detect visible objects regardless of their position. The scale method simulates objects at different distances from the camera by scaling the image according to a gain factor. The fliplr method horizontally flips the image from left to right with a specified probability, increasing the diversity of the dataset and enhancing the learning of symmetrical objects. Lastly, the mosaic method combines four training images into one, simulating different scene compositions and object interactions, which strengthens the understanding of complex scenes.

Through these data augmentation techniques, our model is exposed to a more diverse range of samples during training, enabling it to learn more essential and generalized features, thereby achieving higher detection accuracy. The introduced noise and variations allow the model to learn how to cope with various potential interference factors during training, improving its robustness and stability in practical applications.

Dataset

In the experiment, we chose two parts of RDD2022 named RDD-China, a large open-source dataset containing road surface pictures of many countries, as our experimental dataset. The dataset contains a total of 4,378 pictures, of which 2,401 are taken by drones and 1,977 are taken by on-board cameras. As shown in Fig. 7, there are five types of road damage in the picture: longitudinal cracks (D00), transverse cracks (D10), alligator cracks (D20), potholes (D40), and road repairs. These images are divided into train_set, valid_set, and test_set in an 8:1:1 ratio.

Figure 7 Different types of road damage in the dataset.

Evaluation metrics

In this article, four evaluation criterions, namely, F1-score, mean average precision (mAP), number of parameters (Params), giga floating point operations per second (GFLOPs) and frames per second (FPS) are employed to assess the performances of OBC-YOLOv8 and comparative methods. In detail, correct rate and recall rate are taken as the basic indicators, F1-score and mAP@0.5 calculated according to correct rate and recall rate are taken as the final evaluation indicators to measure the recognition accuracy rate of the model. GFLOPs are used to measure the complexity of a model or algorithm, while Params represent the size of the model. FPS is used to evaluate the speed of detection.

Precision (P) is the ratio of how many true positive samples are among the samples that are predicted as positive samples while recall (R) is the ratio of how many true positive samples are correctly detected among all positive samples. The F1-score is the overall performance of a more comprehensive reaction network considering the influence of P and R. Precision, recall and F1-score are provided by:

(12) P=TPTP+FP,R=TPTP+FN.

(13) F1=2∗P∗RP+R

The mAP is the mean of the average precision (AP) of each category. The mAP calculation formula is as follows:

(14) mAP=1N∑i=1NAPi

where AP is the area enclosed by the P-R curve and the X-axis and Y-axis. The larger the value of mAP index, the higher the accuracy of the model in the recognition task.

The effects of different size of YOLOv8

In the task of road damage detection, accuracy and real-time detection are essential. In order to explore the most suitable model, we conducted comparative experiments on YOLOv8 models of different sizes, and the results are shown in Table 3. (Models of −l and −x size cannot be run due to insufficient GPU memory. However, from the perspective of resource utilization, these two models are not suitable for our real-time road damage detection tasks.).

Table 3 The performance of model with different size.

Models	mAP@0.5/%	Para/106	GFLOPs	F1-score/%	FPS	
YOLOv8n	84.2	3.0	8.1	80.3	243	
YOLOv8s	85.1	11.1	28.4	80.8	164	
YOLOv8m	85.7	25.8	78.7	81.2	102	
OBC-YOLOv8n	86	3.2	8.0	81.9	222	
OBC-YOLOv8s	86.2	12.1	28.6	81.8	151	
OBC-YOLOv8m	86.4	27.1	78.2	82.1	85	

From the aforementioned results, it is evident that each increment in the model size is accompanied by a substantial, multi-fold increase in the number of parameters, yet the gains in accuracy are relatively modest. Notably, the OBC-YOLOv8n model achieves an accuracy level comparable to that of YOLOv8m, despite possessing only one-ninth of the parameters. This underscores the efficiency and effectiveness of choosing the YOLOv8n model as a baseline in terms of balancing performance and computational cost.

The effects of different attention mechanisms

Based on the YOLOv8n model, we chose four attention mechanisms, SE, CBAM, efficient multi-scale attention (EMA) (Ouyang et al., 2023) and CA, to explore their influences on detection accuracy. The experimental results are shown in Table 4. It is not difficult to see from the table that after adding different attention mechanisms, the number of parameters and calculation speed of each method are basically the same, and there is no significant change. With the exception of EMA, which experiences a slight decrease in mAP@0.5, all other attention mechanisms improve to varying degrees. Notably, CA achieves the most significant enhancement, registering a 1.1% increase compared to the baseline. SE also showes a 0.9% improvement on mAP@0.5, but it is slightly below the baseline on mAP@0.5–0.95, indicating poor accuracy at higher thresholds. CA achieves the second-best improvement in this respect, which intuitively illustrates the effectiveness of CA. At the same time, we find that the F1-score index of CA decreased by 0.5%, mainly because its accuracy rate P increased but the recall rate R decreased, indicating that the model with the addition of CA module also had shortcomings. Although the accuracy of prediction is greatly improved, some positive samples would be wrongly classified as negative samples.

Table 4 Performance of different attention modules.

Models	mAP@0.5/%	mAP@0.5−0.95/%	Para/106	GFLOPs	F1-score/%	
YOLOv8n	84.2	57.6	3.0	8.1	80.3	
+EMA	84	58	3.0	8.1	80.4	
+CBAM	84.6	58.5	3.0	8.1	81.2	
+SE	85.1	57.5	3.0	8.1	80.9	
+CA	85.3	58.2	3.0	8.1	79.8	

The effects of different insert layers

CA is introduced into the Neck part to enhance the feature information in the feature map and improve the detection accuracy. In order to explore the influence of different insertion layers on the accuracy of the model, the following experiments are designed: an insertion position is set behind each C2f module in the Neck part and is named L1, L2, L3 and L4 respectively. Based on this, different combinations were made and the corresponding results are shown in Table 5.

Table 5 The effect of different insertion layers of CA module.

L1	L2	L3	L4	P/%	P/%	mAP@0.5/%	mAP@0.5−0.95/%	
				82.6	80.4	85.5	57.4	
✓				82.5	79.4	85.2	58.2	
	✓			85.1	78.8	85.9	59.3	
		✓		79.7	78.8	83	56.2	
			✓	84.6	78.9	85.7	58.9	
✓	✓			81.1	79.7	85.1	57.8	
✓		✓		84.3	79	85.4	58.9	
✓			✓	81.4	81.4	86	59.3	
	✓	✓		82.8	81.1	85.7	58.7	
	✓		✓	85	80.4	85	58.9	
		✓	✓	82.9	81.1	86	58.8	
✓	✓	✓		83.7	85	85.5	58.6	
✓	✓		✓	81.4	81.9	85.5	59	
✓		✓	✓	84.5	77.3	85.4	59.4	
	✓	✓	✓	83.5	77.7	85	58.2	
✓	✓	✓	✓	82.1	79.2	85.2	59.1	

Firstly, we introduce CA modules individually behind each C2f module to assess their individual contributions towards enhancing model performance. The corresponding experimental outcomes are presented in rows 2 to 5 of Table 5. Notably, we observed that the influence of a solitary module on the overall model is unpredictable. Specifically, when L2 and L4 are incorporated, mAP@0.5 witnessed an improvement of 0.4% and 0.2%, respectively. Conversely, the insertion of L1 and L3 does not lead to any improvement in mAP@0.5, but rather a decline. Notably, the worst performance was observed when L3 was integrated, resulting in a 2.5% reduction in mAP@0.5. These findings indicate that the CA module can effectively enhance the detection accuracy of the model in feature maps of small and large size.

To further explore its potential, we proceed to increase the number of CA blocks by adopting different combinations. As evident from the data presented in Table 5, both insertion methods of L1 + L4 and L3 + L4 yield mAP@0.5 values around 86%. However, a closer comparison reveals that the L3 + L4 combination performs better in terms of precision (P) and recall (R). Specifically, compared to the baseline, the L3 + L4 approach improves precision by 0.3%, recall by 0.7%, mAP@0.5 by 0.5%, and mAP@0.5–0.95 by a notable 1.2%. This experiment demonstrates that inserting the CA block at L3 and L4 is beneficial for extracting image feature information more comprehensively, thereby enhancing the detection accuracy of road damage targets. Consequently, we select this insertion scheme as the optimal approach for this model.

Ablation experiment

In order to verify the influence of each module on the detection performance of the improved model, we conducted ablation experiments using YOLOv8n as a baseline in this article. We investigated the role of different modules by combining them on the baseline (the “✓” indicates that the corresponding module is used). Based on Table 6, it is not difficult to see that the module, that has achieved the highest improvement in mAP@0.5 is BoTNet, which has achieved 1.3% improvement. However, due to its self-attentive mechanism, it inevitably introduces a portion of parameters and computations. Then the inclusion of CA module results in a 1.1% improvement with minimal impact on parameters and computation. The addition of the ODConv module allows the model to achieve a 0.9% improvement in mAP while providing the largest improvement in F1-score. In general, a small number of parameters are introduced when the three modules are combined and the speed of the model is not greatly affected. OBC-YOLOv8 not only improves mAP by 1.8%, but also offsets the influence of CA module and improves F1-score by 1.6%. The results validate the effectiveness of OBC-YOLOv8 on road damage detection. As depicted in Fig. 8, the accuracy of OBC-YOLOv8 across all damage classes surpasses that of YOLOv8n. Figure 9 illustrates the performance comparison between YOLOv8n and OBC-YOLOv8 in terms of the indicators mAP@0.5.

Table 6 Module ablation experiments.

ODConv	BoTNet	CA	mAP@0.5/%	P/%	GFLOPs	F1-score/%	
			84.2	3.0	8.1	80.3	
✓			85.1	3.0	7.9	80.8	
	✓		85.4	3.2	8.2	80.5	
		✓	85.3	3.0	8.1	79.8	
✓	✓		85.6	3.2	8.0	81.5	
✓		✓	85.4	3.0	7.9	80.6	
	✓	✓	84.7	3.2	8.3	80	
✓	✓	✓	86	3.2	8.0	81.9	

Figure 8 The performance on different damage types of YOLOv8n and OBC-YOLOv8.

Figure 9 The performance on mAP@0.5 of YOLOv8n and OBC-YOLOv8.

Efficiency experiment

In order to further verify the superiority of OBC-YOLOv8 in road damage detection tasks, we compared the model with other typical methods on RDD-China and the experimental results are shown in Table 7. As evident from Table 7, the OBC-YOLOv8 model significantly outperforms both Faster R-CNN and Cascade-RCNN in terms of mAP@0.5. Specifically, it achieves an increase of 12.8% and 31.2% in mAP@0.5 compared to these respective models. Furthermore, the F1-score also witnesses a substantial improvement, rising by 21.9% and 25.5%, respectively. Similarly, when compared to SSD and EfficientDet, the OBC-YOLOv8 model demonstrates a substantial enhancement in mAP@0.5, achieving an increase of 13.3% and 36.2%, respectively. Concurrently, the F1-score also experiences a noteworthy uplift, rising by 21.9% and 35.9%, respectively. These results clearly demonstrate the superior performance of OBC-YOLOv8 in road damage detection tasks. What’s more, OBC-YOLOv8 has less parameters and higher FPS value. Then we selected YOLOv3-tiny, YOLOv5n and YOLOv5s for comparison. Evidently, the OBC-YOLOv8 model expertly balances detection speed and accuracy, achieving notable improvements in the mAP@0.5 index in different degrees. Specifically, it boasts an increase of 9.9%, 5.4%, and 0.8%, respectively, demonstrating its superior performance in object detection tasks. Although YOLOv5s is 1% better than YOLOv8n in terms of accuracy, the number of parameters it possesses and the computational complexity are huge.

Table 7 Comparison with other methods.

Models	mAP@0.5/%	AR	Para/106	GFLOPs	F1-score/%	FPS	
Faster R-CNN	73.2	52.1	28.3	940.9	60	11	
Cascade R-CNN	54.8	61.2	87.9	110.6	56.4	24	
SSD	72.7	54	26.2	62.7	61	76	
EfficientDet	49.8	44.8	3.8	5.2	46	27	
YOLOv3-tiny	76.1	68.5	8.7	13.0	71.2	220	
YOLOv5n	80.6	76.1	1.7	4.2	77.1	218	
YOLOv5s	85.2	82.6	7.02	15.8	83	156	
YOLOv8n	84.2	77.9	3.0	8.1	80.3	243	
YOLOv8n-BIFPN	84.5	80.5	3.0	8.1	81.3	238	
YOLOv8n-GhostConv	83.6	81.4	2.8	7.7	80.5	256	
YOLOv8n-CondConv	84.9	78.5	3.0	7.8	81.4	231	
YOLOv8n-DWConv	85.6	78	3.0	7.9	81.8	237	
OBC-YOLOv8	86	81.1	3.2	8.0	81.9	222	

We also chose some state-of-the-art modules based on YOLOv8n to verify the performance of our model, such as the BiFPN (Tan, Pang & Le, 2020), GhostConv (Han et al., 2020), CondConv (Yang et al., 2019), and DWConv (Howard et al., 2017). The proposed OBC-YOLOv8 model provides substantial enhancement in terms of accuracy compared to other models. However, these enhancements are accompanied by an increase in inference time. Although our model does not have much computational complexity compared to YOLOv8n, the addition of BoTNet module inevitably increases the number of parameters, which makes a gap between our model and GhostConv. Nevertheless, along with a little bit of parameter count and computations, our method exceeded these methods in mAP@0.5 and F1-score, with 1.5%, 2.4%, 1.1% and 0.4% improvement in mAP@0.5 and 0.6%, 1.4%, 0.5% and 0.1% improvement in F1-score, respectively. Since only some module changes are involved, the number of parameters and FPS do not differ much. Figure 10 displays the Precision-Recall (P-R) curves for both YOLOv8n and OBC-YOLOv8, respectively. It is evident that OBC-YOLOv8 outperforms YOLOv8n in terms of accuracy across all categories. This indicates that the modifications and enhancements incorporated into OBC-YOLOv8n have significantly improved its ability to detect road damages.

Figure 10 The performance on P-R curve of YOLOv8n and OBC-YOLOv8.

The above experimental results show that compared with other promising detection models, OBC-YOLOv8 not only achieves high detection accuracy, but also has high detection efficiency.

Generalization experiment

To verify the generalization ability of the model, we conducted experiments on the RDD-Japan dataset, which includes 8,586 images of seven categories of damage, they are longitudinal cracks (D00), transverse cracks (D10), alligator cracks (D20), potholes (D40), damaged crosswalk (D43), damaged paint (D44) and manhole cover (D50).

The experimental results are depicted in Fig. 11, revealing a compelling observation. It is evident that our model outperforms the baseline in terms of detection accuracy across all categories of road damage. Notably, when compared to its performance on the RDD-China dataset, both the baseline and our model exhibit a considerable decline in accuracy when evaluated on the RDD-Japan dataset. This disparity stems from the diverse weather and lighting conditions present in the RDD-Japan images, which have a profound impact on the visibility and contrast of pavement damages. Variations in weather can introduce shadows, reflections, and noise into the imagery, significantly degrading image quality and escalating the challenge of detection. Nevertheless, under these circumstances, the OBC modules play an important role in helping the model pay more attention to the road damage features, which makes our model achieve a noteworthy 1.5% enhancement in the mAP@0.5 metric, underscoring its robust generalization capabilities.

Figure 11 The performance on mAP@0.5 of YOLOv8n and OBC-YOLOv8 on RDD-Japan.

Visualization analysis

In order to sufficiently verify the superiority of OBC-YOLOv8, some images are selected in the dataset for comparative testing to evaluate the model intuitively. The comparison results are shown in Fig. 12. It is not difficult to see that the confidence degree of detection targets in YOLOv8n is lower than the OBC-YOLOv8. There are also some cases of missing detection. For example, in the first comparison, YOLOv8n misses the cracks of type D00, and in the third comparison, the cracks of type D10 can be identified by OBC-YOLOv8 while is missed by YOLOv8n. However, there are also some defects. In RDD-China, because there is no category of potholes in the training, the model will identify manhole covers as repairs during the forecasting process. In RDD-Japan, some images with complex shadows will affect the model’s detection and inference confidence for some cracks and potholes. The test results show that while there are still a few defects, OBC-YOLOv8 offers fewer missed and false checks compared to YOLOv8n.

Figure 12 The detection results of YOLOv8n (down) and OBC-YOLOv8 (up).

Conclusion

In this article, a novel road damage detection model named OBC-YOLOv8 is proposed. OBC-YOLOv8 firstly leverages ODConv block to capture the complex and diverse features of road damages and then utilize BoTNet to comprehensively extract the global and local feature information. Finally, CA is introduced to reduce the interference of irrelevant features and significantly improve the detection accuracy. Extensive experimental results on RDD-China dataset validate the effectiveness and efficiency of OBC-YOLOv8.

In the future, we are devoted to enhance the performance of OBC-YOLOv8 further and simultaneously make lightweight research. Specifically, knowledge distillation, channel pruning and weight pruning will be leveraged to improve the accuracy of the model further and compress the model to facilitate the deployment on different terminal devices.

Supplemental Information

Supplemental Information 1 The code of OBC-YOLOv8.

Additional Information and Declarations

Competing Interests

Author Contributions

Data Availability

The authors declare that they have no competing interests.

Shizheng Zhang conceived and designed the experiments, prepared figures and/or tables, authored or reviewed drafts of the article, and approved the final draft.

Zhihao Liu conceived and designed the experiments, performed the experiments, performed the computation work, prepared figures and/or tables, and approved the final draft.

Kunpeng Wang conceived and designed the experiments, performed the experiments, analyzed the data, performed the computation work, prepared figures and/or tables, and approved the final draft.

Wanwei Huang analyzed the data, authored or reviewed drafts of the article, and approved the final draft.

Pu Li analyzed the data, authored or reviewed drafts of the article, and approved the final draft.

The following information was supplied regarding data availability:

The code is available at GitHub and Zenodo:

- https://github.com/wulihuge/OBC-YOLOv8

- wulihuge. (2024). wulihuge/OBC-YOLOv8: readme (OBC-YOLOv8). Zenodo. https://doi.org/10.5281/zenodo.13912921.

The dataset is available at figshare: Arya, Deeksha; Maeda, Hiroya; Sekimoto, Yoshihide; Omata, Hiroshi; Ghosh, Sanjay Kumar; Toshniwal, Durga; et al. (2022). RDD2022-The multi-national Road Damage Dataset released through CRDDC'2022. figshare. Dataset. https://doi.org/10.6084/m9.figshare.21431547.v1.

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
