# Peer review of "OBC-YOLOv8: an improved road damage detection model based on YOLOv8"

_PeerJ Computer Science, doi:10.7717/peerj-cs.2593_

## Round 0.1 · original submission · Major Revisions

Please revise the paper according to the reviewer's comments.

Reviewer 1 ·

Basic reporting

I have included a thorough manuscript review in the additional comments section. The objective questions asked here are also addressed in the additional comments.

Experimental design

I have included a thorough manuscript review in the additional comments section. The objective questions asked here are also addressed in the additional comments.

Validity of the findings

I have included a thorough manuscript review in the additional comments section. The objective questions asked here are also addressed in the additional comments.

Additional comments

The manuscript introduces a new model called OBC-YOLOv8 for detecting road damage. This model incorporates ODConv blocks, BoTNet, and the CA mechanism to improve the accuracy of detection.

Benefits:

1. The incorporation of ODConv, BoTNet, and CA processes into YOLOv8 is an innovative strategy that significantly improves the extraction of features and the accuracy of detection.
2. The model undergoes a rigorous evaluation using many metrics (F1-score, mAP, Params, GFLOPs) and on a diversified dataset (RDD2022-China).
3. The attention mechanism of the CA effectively reduces irrelevant features and enhances detection accuracy, as proven.
BoTNet's integration enhances the extraction of both global and local features, overcoming the drawbacks of relying simply on CNNs or Transformers.
5. The text demonstrates substantial enhancements in mAP50 and F1-score when compared to baseline models, showing the efficacy of the proposed approach.

Drawbacks:

1. The inclusion of ODConv and BoTNet modules amplifies the intricacy of the model and incurs higher computing expenses.
2. The time it takes to make inferences has increased, which could have an impact on applications that require real-time processing.
3. The manuscript mainly contrasts OBC-YOLOv8 with YOLOv8, Faster R-CNN, and SSD. Enhancing the claims could be achieved by incorporating a wider range of baseline data.
4. The utilization of RDD2022-China restricts the applicability of the findings. Conducting tests on supplementary datasets would yield a more thorough assessment.
5. Although the impacts of specific modules are addressed, doing more comprehensive ablation experiments could offer a more profound understanding of their respective contributions.

Concerns:
1. The publication lacks discussion on the model's ability to generalize effectively to diverse road conditions or environments that are not included in the RDD2022-China dataset.
2. The potential obstacles and solutions for implementing OBC-YOLOv8 in real-world situations are not addressed.
3. The manuscript lacks information regarding the computational resources needed for training and inference, which is crucial for practical implementations.
4. The discussion does not address the influence of data augmentation approaches on enhancing the model's performance.

Suggestions:

1. Elaborate on the compromises that arise when aiming for enhanced precision while also facing higher computing demands. Please provide a comprehensive analysis of the inference times and computing costs associated with various configurations.

2. Incorporate comparisons with supplementary cutting-edge models for road damage detection, such as EfficientDet, CornerNet, or other advanced one-stage and two-stage detectors.

3. Evaluate the model's performance on supplementary datasets, encompassing diverse countries and variable road conditions, to verify its ability to generalize.

4. Perform more extensive ablation research to separate and analyze the individual contributions of each module (ODConv, BoTNet, CA) to the overall improvement in performance. The report should contain comprehensive metrics for each ablation procedure.

5. Examine the possible difficulties in implementing OBC-YOLOv8 in real-world scenarios, including managing diverse lighting situations, weather fluctuations, and the need for real-time processing.

6. Provide explicit information about the computing resources utilized during the training and inference processes, including specifics on the GPU specifications and the duration of the training. This will offer valuable insights into the model's potential to scale and its practicality for deployment.

7. Improve the visualization section by include additional examples that illustrate the disparities in detection accuracy between YOLOv8 and OBC-YOLOv8. Identify particular instances where OBC-YOLOv8 demonstrates exceptional performance or shortcomings, offering valuable observations into its advantages and constraints.

8. Add a section that explores the significance of data augmentation in model training. Provide a comprehensive explanation of the specific techniques employed and their influence on the model's performance.

By considering these aspects, the paper will present a more thorough and strong assessment of the proposed OBC-YOLOv8 model, providing valuable perspectives on its practical uses and constraints.

Cite this review as

Reviewer 2 ·

Basic reporting

It's a bit good article.
Structured organized very well and formatted as per legitimate requirements.

Experimental design

Seems to be original and adding new value to the knowledge areas.

Validity of the findings

It has readership appeal and technical merit
Results are encouraging but there is scope for substantial improvement

Additional comments

References are relevant, good enough, adequate and recent ones too.
Has readership appeal and impact,
Results are encouraging but need to explore further for enriching the merit of the article.
Technical writing is good,
Overall a good contribution to the field.

Cite this review as

Reviewer 3 ·

Basic reporting

All comments have been added in detail to the last section.

Experimental design

All comments have been added in detail to the last section.

Validity of the findings

All comments have been added in detail to the last section.

Additional comments

Review Report for PeerJ Computer Science
(OBC-YOLOv8: An improved road damage detection model based on YOLOv8)

1. Within the scope of the study, a YOLO-based model was proposed to perform road damage detection operations on an open-source dataset shared as open source in 2022.

2. In the introduction, road damage detection technologies and main contributions are mentioned at a basic level. In this section, the difference of the study from the literature and its main contributions to the literature should be explained in more detail.

3. The related works section is very limited. In this section, in order to emphasize the importance of the subject and its place in the literature more clearly, the literature should be analyzed in more detail and the advantages, disadvantages, results, data preprocessing/augmentation should be interpreted in terms of the comparison with the literature within the scope of the proposed model.

4. When the proposed OBC-YOLOv8 model is examined, sections such as network architecture, bottleneck structures, multi-head attention module and the originality points of the model are clearly mentioned. Although there are many YOLO-based models in object detection problems in the literature, there are more up-to-date versions such as YOLOv10. In the scope of this study, it should be explained more clearly why YOLOv8 is specifically taken as the basis within the scope of the proposed model, despite its more up-to-date versions.

5. Within the scope of the study, an open source dataset frequently used in road damage detection problems was preferred. Since a new model is proposed for these detection processes in the study, performing the application on only one dataset limits the usability of the model. Here, it is very important to emphasize the usability of the model by performing object detection processes with at least one of the other open source datasets in the literature that can be used for similar problems in road damage detection.

6. There are very serious deficiencies in the evaluation metrics section. In order to analyze the results correctly in object detection problems, the metrics must be obtained absolutely completely. For this reason, it is very important to obtain all the missing metrics (Average Recall (AR), Optimal Localization Recall Precision (oLRP), Precision-Recall Curve and similar ones) absolutely completely.

7. In order to clarify the superiority of the proposed model more clearly, it is recommended to include the results of both single-stage and two-stage state-of-the-art object detection models based on deep learning in the literature in the comparison section with other models.

In conclusion, although the study proposes a deep learning based model with a certain level of specificity for road damage detection, all the above mentioned parts should be explained step by step and covered completely with updates/additions made where relevant in the paper.

Cite this review as

Reviewer 4 ·

Basic reporting

The article is logically organized, with each section flowing smoothly into the next. The introduction provides a comprehensive background, clearly stating the objectives. Technical details are accurate, and results are presented clearly with appropriate visual aids.

The suggestions for improvment are as follows:
- The English language would be improved if authors pay attention to missing articles in their writing.
- The text has inconsistent references. Authors refer to only surnames, only the first name, or both in their writing. Suggestion for consistency, it will be better if the authors use the authors’ surname only. For example, in lines 54 (“Vung Pham et al.” should be “Pham et al.”), 72 (“Arunabla et al.” should be “Roy and Bhaduri”), and 78 (“jNirmal et al.” should be “Rout et al.”), please see the yellow highlight in the attached file.
- Figures 1, 2, and 6 show enlarged text that becomes blurry and has a faint outline at the edges. It is recommended that text-containing images use vector graphics.
- Another area for improvement is the related work section. While it effectively describes structure and functionality of YOLOv8, it could benefit from a discussion of YOLOv8's application in object detection and road damage detection. This addition will make the article more comprehensive and relevant to the field.
- Please confirm if the last term in equation (1), αW1, is correct or if it is αWn.
- Line 146, the authors define “e represents the multiplication…” but there is no “e” in the equation (1).
- The caption of Figure 2, “the Network structure …” should be “The network structure…”
- The caption of Figures 3 and 6, “the structure …” should be “The structure”
- In Figure 4, it would have been better if the authors had used different colors for the middle box between the ResNet Bottleneck and Bottleneck Transformer.
- Line 190, what is “SE attention”? Is it from SENet?
- After line 206, the paragraph without the line number in the third line, “repectiWely.” should be “, respectively.”
- Figure 8 (a) caption/title of graph is missing.

Experimental design

The explanation of the research problem is distributed across the introduction and methods sections, where it discusses that road damage detection is more challenging than general object detection and addresses the limitations of YOLOv8 in this area. If these issues could be consolidated into a separate section, it would enhance the clarity of the article's structure and make it easier for readers to follow.
In the experiments section, it is mentioned that YOLOv8 is used as a baseline and that the YOLOv8n pre-trained model is employed for comparison in the experimental design. However, YOLOv8 has five pre-trained models: n, s, m, l, and x. While YOLOv8n has the fewest parameters, it also has the lowest performance. The experiments should include the detection results for all five pre-trained models or at least use the one with the best performance for comparison.
In the Ablation Experiment section, the combination of the ODConv and CA modules is not included. Although the paper clearly states that BoTNet provides the most significant improvement in overall detection performance, the experimental design should be more comprehensive.
In the Experimental Environment section, only the operating system, CPU, and GPU are mentioned, but parameters such as learning rate, momentum, dropout, batch size, and number of epochs are not specified. This lack of detail makes it difficult for readers to replicate the experiments.

Validity of the findings

The paper builds upon YOLOv8 by replacing and adding ODConv, BoTNet, and CA modules, resulting in a modest performance improvement with 1.8% and 1.6% increases in mAP50 and F1-Score, respectively. However, the improvements are relatively small, and these modules are existing research contributions without significant modifications, which may indicate limited innovation.
The experimental dataset for this paper comes from the Crowdsensing-based Road Damage Detection Challenge Competition, which provides over 40,000 training images from six countries. However, the authors only selected 4,378 images from this dataset and did not specify the number of images used for training, validation, and testing. This approach could undermine the credibility of the research findings.

Annotated reviews are not available for download in order to protect the identity of reviewers who chose to remain anonymous.
Cite this review as

·

Basic reporting

No Comment

Experimental design

No Comment

Validity of the findings

No Comment

Additional comments

Road damage detection is crucial for maintaining infrastructure and ensuring road safety. Timely and accurate detection of road damage can prevent accidents, reduce repair costs, and improve the overall quality of transportation networks. YOLOv8 represents the latest advancements in the You Only Look Once (YOLO) family of object detection models. It incorporates cutting-edge techniques in deep learning and computer vision, offering improved accuracy and speed over its predecessors.
With the increasing availability of high-quality road imagery datasets and advancements in data collection methods (e.g., drones, vehicle-mounted cameras), researchers have the resources needed to train and validate their models effectively.

Cite this review as

---

## Round 0.2 · Minor Revisions

Please revise the paper according to the reviewer's final comments.

Reviewer 1 ·

Basic reporting

NA

Experimental design

NA

Validity of the findings

NA

Additional comments

NA

Cite this review as

Reviewer 3 ·

Basic reporting

All comments have been added in detail to the last section.

Experimental design

All comments have been added in detail to the last section.

Validity of the findings

All comments have been added in detail to the last section.

Additional comments

Review Report for PeerJ Computer Science
(OBC-YOLOv8: An improved road damage detection model based on YOLOv8)

Both the responses to the reviewer comments and the changes made to the paper were examined in detail. Although some of the responses and changes made were very limited, the overall state of the paper is at an acceptable level.

Cite this review as

Reviewer 4 ·

Basic reporting

The paper is well-organized, with each section transitioning smoothly to the next. The introduction offers a thorough background and clearly states the objectives. The technical details are presented with precision, instilling confidence in the research's accuracy. Results are presented clearly, accompanied by appropriate visual aids.

Experimental design

The authors have implemented the recommended modifications and enhancements.

Validity of the findings

This paper has demonstrated a greater level of impact and has yielded superior results.

Additional comments

This paper meets the criteria for publication in this journal and is recommended for acceptance. The authors have made commendable efforts to modify and enhance the suggestions. Despite these efforts, several consistent issues remain. For instance:
- The captions for Figures 8, 9, and 10 refer to YOLOv8n, while the accompanying text mentions YOLOv8. This inconsistency may confuse readers. Since the authors clarified that YOLOv8n was chosen for this study, it should consistently reference YOLOv8n throughout.
- On page 4, lines 152, 165, and 177, the term "Fig X" is used to reference Figure X, whereas on page 14, lines 367 and 368, the full term "Figure X" is employed. This inconsistency should be addressed.
- There are discrepancies in the spacing of parentheses; some have spaces before and after while others do not. This occurs, for example, on page 4, line 172; page 5, line 189; and page 7, line 228.
It would be advisable for the authors to thoroughly review and proofread the text for these inconsistencies.

Cite this review as

---

## Round 0.3 · accepted · Accept

According to the comments of reviewers, after comprehensive consideration, it is decided to accept it.